# Endoglin Expression and Microvessel Density as Prognostic Factors in Pediatric Rhabdomyosarcoma

**DOI:** 10.3390/jcm10030512

**Published:** 2021-02-01

**Authors:** Joanna Radzikowska, Antoni Krzeski, Anna M. Czarnecka, Teresa Klepacka, Magdalena Rychlowska-Pruszynska, Anna Raciborska, Bozenna Dembowska-Baginska, Maciej Pronicki, Andrzej Kukwa, Janusz Sierdzinski, Wojciech Kukwa

**Affiliations:** 1Department of Otorhinolaryngology, Faculty of Dental Medicine, Medical University of Warsaw, 19/25 Stepinska St., 00-739 Warsaw, Poland; jwronska@poczta.onet.pl (J.R.); antoni.krzeski@wum.edu.pl (A.K.); 2Department of Soft Tissue/Bone Sarcoma and Melanoma, Maria Sklodowska-Curie Institute—Oncology Center, 5 Roentgena St., 02-781 Warsaw, Poland; anna.czarnecka@gmail.com; 3Department of Experimental Pharmacology, Mossakowski Medical Research Centre, Polish Academy of Sciences, 5 Pawinskiego St., 02-106 Warsaw, Poland; 4Department of Pathology, Institute of Mother and Child, 17a Kasprzaka St., 01-211 Warsaw, Poland; tklepacka@wp.pl; 5Department of Oncology and Surgical Oncology for Children and Youth, Institute of Mother and Child, 17a Kasprzaka St., 01-211 Warsaw, Poland; magdarychlowska@wp.pl (M.R.-P.); anna.raciborska@hoga.pl (A.R.); 6Department of Pediatric Oncology, The Children’s Memorial Health Institute, 20 Dzieci Polskich St., 04-730 Warsaw, Poland; b.dembowska@ipczd.pl; 7Department of Pathology, The Children’s Memorial Health Institute, 20 Dzieci Polskich St., 04-730 Warsaw, Poland; m.pronicki@ipczd.pl; 8Department of Otolaryngology and Head and Neck Diseases, School of Medicine, University of Warmia and Mazury, 30 Warszawska St., 10-082 Olsztyn, Poland; andrzejkukwa41@gmail.com; 9Department of Medical Informatics and Telemedicine, Medical University of Warsaw, 14/16 Litewska St., 00-581 Warsaw, Poland; janusz.sierdzinski@wum.edu.pl

**Keywords:** endoglin, microvessel density, rhabdomyosarcoma, soft tissue sarcoma, neoangiogenesis

## Abstract

(1) Background: The study proposed to analyze microvessel density (MVD) in rhabdomyosarcoma (RMS) based on the expression of angiogenesis markers and define its prognostic role in this group of patients. (2) Methods: The study included forty-nine pediatric patients diagnosed with RMS. Tumor tissue expression of CD31, CD34, and CD105 was analyzed. MVD was calculated and correlated with clinical RMS prognostic parameters. (3) Results: CD31, CD34, and CD105 are expressed in all RMS cases. MVD/CD105 was significantly higher in the RMS group than in the control group. The mean and median values of MVD/CD105 in RMS were lower than MVD/CD31 and MVD/CD34. MVD/CD105 was significantly higher in patients with alveolar RMS and those with metastatic disease. Patients with higher levels of MVD/CD105 had a higher risk of death (HR = 1.009). (4) Conclusion: CD105 is a relevant angiogenesis marker in pediatric RMS, and MVD/CD105 is an independent risk factor of short overall survival in children with RMS.

## 1. Introduction

Rhabdomyosarcoma (RMS) is the most common soft tissue sarcoma in children, accounting for about 4.5% of all malignant tumors [1]. In the USA, 350 new RMS cases are diagnosed each year in patients under 20 years of age [1,2]. According to Polish statistics, rhabdomyosarcoma accounts for 3.8% of all malignancies in children [3]. The first peak incidence occurs between 2 and 6 years of age, the second between 10 and 18 [4]. Histologically, RMS is divided into two major subtypes, differing in their molecular genetics and prognosis. The embryonal subtype (ERMS) represents about 75% of cases, typically in the younger age group, located in the head and neck region or the genitourinary tract. The alveolar subtype (ARMS), found in 25% of cases, occurs more often in older children, typically in the trunk and extremities [4]. The pathogenesis of RMS is still not fully understood. Disruptions in muscle progenitor cells’ growth and differentiation processes are believed to play a role in RMS’s malignant transformation and progression [4]. Various epigenetic changes are found in ERMS tumors, including loss of heterozygosity in the 11p15 locus (insulin-like growth factor IGF-2) as well as trisomy of chromosome 8, mutations of the TP53 gene, and deregulations of the p38, JKN, ERK, cyclin, and cyclin-dependent kinases activity [4,5,6,7,8,9,10]. Pathognomonic translocations are known for ARMS. The most common translocation, t(2;3) (q35;q14), gives rise to the chimeric protein PAX3-FKHR, which affects the growth, mobility, differentiation, and apoptosis of tumor cells, intensifying carcinogenesis. ARMS cases with this translocation, termed fusion gene-positive alveolar RMS (ARMSp), are known for a worse prognosis [4,11]. A genetic profile may become a significant biomarker in therapeutic protocol selection in the future. Currently, it is known that RMS prognosis depends on multiple factors. Favorable prognostic factors are: the embryonal histological subtype; fusion gene-negative status; primary localization in the orbit or other areas of the head and neck (except for the parameningeal region) as well as the genitourinary tract (except for the bladder and prostate); lack of distant metastases at the time of diagnosis; R0 tumor excision; tumor size ≤5 cm; age <12 years; and long time to relapse [4,12]. After diagnosis, the five-year survival rate in children with a localized disease who receive combined treatment is over 70% [4]. Children with metastatic RMS at diagnosis have the worst prognosis: 25% disease-free survival (DFS) at three years [4]. To improve the failure-free survival (FFS) in this patient group, researchers and clinicians are currently focusing on developing molecularly targeted (personalized) treatment. Novel treatment targets for RMS are needed.

Assessment of the degree of vascularization of tumors and analysis of angiogenesis-related proteins may allow candidate patients to select antiangiogenic therapies [13]. Microvascular density (MVD) is a widely used quantitative assessment method for the density of neoplastic blood vessels in tumor tissue. It serves as a useful marker of neoangiogenesis intensity [14]. The most commonly used markers of vascular endothelial cells include proteins CD31, CD34, and factor VIII [14] and endoglin (CD105), which is expressed by the proliferating cells of newly forming blood vessels. In fact, the prognostic value of endoglin expression has also been confirmed for multiple solid tumors, including selected sarcoma cohorts [15]. Therefore endoglin is considered a specific marker of tumor vasculature and a potential target for an antiangiogenic therapy [16]. Increased MVD, measured based on endoglin expression (referred as to MVD/CD105), has been correlated with a worse prognosis for breast, lung, prostate, colorectal, ovarian, stomach, liver, gastrointestinal, head, neck, and kidney cancers [17,18,19,20,21,22,23,24,25,26]. The use of monoclonal antibodies against endoglin in oncology treatment is currently under investigation. It is expected that combining anti-CD105 antibodies with VEGF inhibitors may improve the effectiveness of antiangiogenic therapy [15]. Although TRC105 (anti-CD105) treatment was initially indicated as potentially active in sarcomas [27], it did not demonstrate clinically significant efficacy in patients with advanced or metastatic angiosarcoma. In fact, a Phase 3 TAPPAS trial evaluating TRC105 combined with pazopanib in patients with advanced or metastatic angiosarcoma was terminated for efficacy data from more than 120 patients. Nevertheless, at the same time, preclinical experiments have revealed that monoclonal antibody-drug conjugates against CD105 and nigrin-b A or cytolysin are effective in animal models of Ewing sarcoma [28]. Stratification of OS and/or PFS by sarcoma endoglin expression may direct future treatments in other STS histologies. Targeted therapy directed against endoglin may potentially be considered in selected soft tissue sarcomas, where increased endoglin expression has been reported [15,29]. Although the first report has shown CD105 and CD31 expression in RMS, its impact on overall survival (OS) was not defined and indicated as a candidate for further research [13].

This study aimed to analyze the expression of endothelial markers CD105, CD31, and CD34 in pediatric rhabdomyosarcoma. The secondary aim of the study was to define MVD prognostic value and correlate MVD with clinical and pathomorphological parameters of RMS.

## 2. Materials and Methods

### 2.1. Study Population

Forty-nine participants with RMS treated between 2000 and 2016 at the Department of Oncology and Surgical Oncology for Children and Youth of the Institute of Mother and Child in Warsaw (Poland) and the Department of Pediatric Oncology of The Children’s Memorial Health Institute in Warsaw, Poland were enrolled in the study. Patients under 18 years at the time of diagnosis were included. Medical history analysis covered primary tumor stage, treatment, and follow-up data. Patients who developed another malignancy during the five-year observation period were excluded from the study. Primary tumor tissue was collected in treatment-naïve cases during biopsy or surgery before chemo- and/or radiotherapy. The quality of formalin-fixed paraffin-embedded primary tumor sections was confirmed, and the amount of tumor tissue in specimens was analyzed.

Normal striated muscle tissue from sarcoma-free individuals was used as control. The control group consisted of 18 participants under 18 years of age who had undergone tonsillectomy for sleep-disordered breathing or a thyroglossal duct cyst excision between 2005 and 2015.

The project was approved by the local bioethics committee of the Medical University of Warsaw.

### 2.2. Analyzed Clinical Parameters

The following clinical and pathomorphological parameters were selected for statistical analysis: age at diagnosis (three age groups: <1 y, 1–9 y, ≥10 y); sex (male or female); primary tumor location—favorable (orbit, head, and neck regions, excluding the parameningeal area; genitourinary tract, excluding the bladder and prostate) versus unfavorable (parameningeal area, extremities, genitourinary tract occupying the bladder and prostate gland, and other areas); histopathological subtype—favorable (ERMS) versus unfavorable (containing ARMS tissue); T trait (T1 versus T2); metastasis to regional lymph nodes (N0 versus N1); the presence of distant metastases (M0 versus M1); tumor size (a, ≤5 cm versus b, >5 cm); and disease stage based on the TNM classification for RMS, as defined by Intergroup Rhabdomyosarcoma Study Group [30]. Overall survival (OS) analysis employed an observation period of at least five years from diagnosis. For OS analysis, patients with a follow-up shorter than five years were excluded.

### 2.3. Immunohistochemistry

Slices of 3 μm were analyzed. For CD31 and CD34 analysis, dewaxing, hydration, and thermal unmasking of the antigen were conducted with the DAKO PT Link module and with a high-pH solution (EnVision^TM^FLEX Target Retrieval Solution, High pH, K8004). For CD105 analysis, which required enzymatic digestion to expose antigenic determinants, dewaxing, hydration, and unmasking were done manually, and proteinase K (DAKO, S3020) was used (5 min, room temperature). The following primary antibodies were used: anti-CD31 (clone JC70A, Ready-to-Use, DAKO IR610), anti-CD34 (clone QBEnd 10, Ready-to-Use, DAKO IR632), and anti-CD105 (clone SN6h, 1:20, DAKO M3527). EnVision^TM^ FLEX/HRP (DAKO SM802), containing peroxidase and secondary antibodies were used for detection. Diaminobenzidine (EnVision^TM^ FLEX DAB+ Chromogen, DAKO DM827) was employed for secondary antibody visualization. Contrast staining with hematoxylin was employed.

### 2.4. Microvessel Density Assessment

A widely used method developed by Weidner was used to calculate microvessel density [31]. The method of MVD assessment developed by Weidner in 1991 employs immunohistochemistry to identify specific vascular endothelial cell markers in the tumor tissue. The MVD is the number of labeled single endothelial cells or clusters, defined as “microvessels,” which are identified in the most active angiogenic regions (hot spots) per square millimeter [31]. MVD was calculated based on the expression of selected endothelial markers. Slides were blinded, thus researchers were not aware of the participants’ clinical status. MVD was assessed by two independent researchers using a NIKON LABOPHOT-2 optical microscope. MVD was assessed based on immunohistochemical staining of CD31 (MVD/CD31), CD34 (MVD/CD34), and CD105 (MVD/CD105), consistent with the method developed by Weidner [31]. Preparations were initially assessed at 100× magnification to select three fields of view with the greatest vascularization (the hot spots). Microvessels in each of the selected fields were then counted at 400 magnification (in 0.196 mm^2^). Every immunopositive structure (endothelial cell or cell cluster) clearly separate from neighboring microvessels, neoplastic cells, or other connective tissue elements was treated as a microvessel, as defined by Weidner [31]. Vessels with visible muscle layers were excluded from analysis as these are not classified as microvessels. The MVD was defined as the mean number of microvessels in the three most vascularized fields of view per 1 mm^2^.

### 2.5. Statistical Analysis

The SAS 9.4 statistical package was used for analysis. The collected data results were subject to descriptive analysis. The Shapiro-Wilk test was used to assess the distribution of selected continuous variables with Gauss’ normal distribution. No normal distribution was found for all tested variables. The assessment of the significance of differences for the tested marker expressions was performed using a U Mann–Whitney nonparametric single-factor test, Kruskal–Wallis variance analysis (ANOVA), and Chi-square tests. Spearman correlation analysis was used to demonstrate the correlation between variables and tested marker expressions. Further evaluation of selected empirical variables was performed using Kaplan–Meier survival analysis, and the multifactor Cox proportional hazard model was used to determine independent prognostic factors influencing survival. Criteria for the selection of variables and their division were based on the ROC curve or parameters of distribution (e.g., averages, medians, etc.). The significance level *p* < 0.05 was used in the analyses. Total survival was defined as the time from RMS diagnosis to death.

## 3. Results

### 3.1. Study Population

The mean age at diagnosis in the study group was 5.76 ± 4.9 years (1 month to 17.9 years). The study group was composed of 19 girls and 30 boys. In 17 cases, primary tumor location was estimated as prognostically favorable and unfavorable in the remaining 32. ERMS was diagnosed in 28 cases and ARMS in 21 cases (Figure 1, Table 1). Fifteen participants were diagnosed with stage 1 disease, 2 participants with stage 2, 15 participants with stage 3, and 17 participants (35% of the study group) were diagnosed with stage 4. Mean follow-up time in the study group was 5.6 ± 4.8 years (6 months to 18.5 years). Twenty-two patients died due to RMS. The mean survival was 33 ± 16 months (from 6 months to 5.75 years).

Twelve girls and six boys were selected for the control group. The mean age in the control group was 11.67 ± 4.87 (Table 2).

### 3.2. CD105, CD34, and CD 31 Expression

Expression of CD105 was primarily observed in immature, small-caliber blood vessels (Figure 2). The median MVD/CD105 in RMS tissue was 107.14 ± 61.36 per 1 mm^2^ and the lowest compared to other markers of angiogenesis. MVD/CD105 in RMS tissue was significantly higher than in the healthy control group. Furthermore, a statistically significant relationship between higher MVD/CD105 and ARMS diagnosis (Z = −2.08, *p* = 0.037) was detected (Figure 3).

Expression of CD31 and CD34 was mostly observed in both small- and large-caliber vessels. The median values of MVD/CD31 and MVD/CD34 in RMS tissue were 142.86 ± 112.69 and 168.37 ± 156.83, respectively. MVD for both CD31 and CD34 was significantly lower in RMS than in the control group (Table 3).

A statistically significant relationship between MVD/CD31, MVD/CD34, and MVD/CD105 was found. The relationship between MVD/CD31 and MVD/CD105 was very strong (Spearman correlation = 0.72), and the correlations between MVD/CD31, MVD/CD34, between MVD/CD34 and MVD/CD105 were also strong (Spearman correlation = 0.56 for both correlations). CD31, CD34, and CD105 expression was also significantly higher in primary tumors of participants with distant metastases at the time of diagnosis (for CD31: Z = −2.36, *p* = 0.018; for CD34: Z = −2.83, *p* = 0.005; for CD105: Z = −2.45, *p* = 0.014) than those with locoregional disease.

### 3.3. Clinical and Pathological Prognostic Factors

Overall survival analysis included 42 out of 49 patients, and the follow-up period was five years. For the remaining seven patients, the follow-up period did not exceed five years; therefore, they were excluded from the analysis. The five-year survival rate was 52% (Figure 3). Kaplan–Meier survival analysis showed significantly lower five-year overall survival of patients with ARMS than those with ERMS (log-rank test = −2.02, *p* = 0.04) (Figure 4), metastasis to regional lymph nodes N1 versus N0 (log-rank test = −2.25, *p* = 0.02), distant metastases M1 versus M0 (log-rank test = −2.74, *p* = 0.006), and an advanced disease stage (Chi-square test = 7.82, *p* < 0.05).

Higher mortality risk was observed in RMS patients with metastasis to regional lymph nodes at the time of diagnosis (hazard ratio = 11.51) and in patients under one year of age and over ten years of age at the time of diagnosis (the hazard ratios were 7.97 and 6.22, respectively). A significantly higher five-year survival was observed in participants aged 1–9 years (Chi-square test = 7.88; *p* = 0.02). In histopathology analysis, increased mortality risk was observed in participants with higher MVD/CD105 (HR = 1.009). For CD105, the size of the area under the ROC curve is AUC = 0.68. Therefore, based on the assumed level α = 0.05 and the obtained value of statistics Z = 2.11 for *p* = 0.035, we conclude that the division of MVD/CD105 using the cut-off value = 124.15 corresponds to the maximum Youden index = 0.37, and it is significantly more advantageous than the random division of patients into two groups for this variable (Figure 5). The Kaplan–Meier survival analysis was performed using the cut-off value = 124.15 for the MVD/CD105. The result of log-rank = 2.29 for *p* < 0.022 confirms significant differences in patient survival for the MVD/CD105 parameter (Figure 6). For MVD/CD31, MVD/CD34, and the remaining clinical and pathological parameters, no significant effect on mortality rate was observed (Table 4).

## 4. Discussion

The development of a network of neoplastic blood vessels within the tumor, essential for its development, directly affects patient prognosis. A correlation of worse survival rates with increased microvascular density was confirmed for multiple cancers as well as melanoma, myeloma, and malignant neoplasms of the central nervous system [32]. At the same time, studies correlating markers of angiogenesis with survival rates in sarcomas are still limited. In particular, expression of CD31, CD34, or CD105 in neoplastic vessels was described in selected cancers [33,34,35] but may not be relevant for sarcomas. The highest value of MVD in epithelial tumors was reported in peripheral regions [36], while in soft tissue sarcomas, blood vessel distribution and MVD are fairly even throughout the whole tumor [37]. Moreover, tumors in children mostly exhibit greater angiogenic potential than tumors in adults [38]. For many cancer types, including lung, stomach, colorectal, ovarian, or prostate cancer, a correlation has been shown between MVD and prognostically significant clinical and pathological factors [39]. Similar reports on sarcomas are limited in number.

In the presented study, MVD was assessed in pediatric RMS based on the expression of CD31, CD34, and CD105 and correlated with selected clinical and pathomorphological factors as well as overall survival. Our results are novel in the sarcoma field. Sarcomas, including RMS, have not been characterized with all three markers before. Kreuter et al. assessed MVD/CD31 in a heterogeneous group of 60 sarcomas, including various histological tumor variants and both primary and metastatic tumors. The median MVD/CD31 calculated with Weidner’s method was 52 in 0.26 mm^2^, which is similar to that presented in our work when data are recalculated for one mm^2^ [40]. In another study of 44 osteosarcomas, 20 chondrosarcomas, and 5 Ewing sarcomas, rich vascularization was shown for MVD/CD34 with Weidner’s method. Median MVD/CD34 was 39.7 ± 31 for osteosarcomas, 44.6 ± 33.8 for Ewing sarcoma cases, and 12.9 ± 32.2 for chondrosarcomas [41]. In terms of prognostic MVD marker significance, Kreuter et al. showed the paradoxically favorable influence of MVD/CD31 in 44 patients with osteosarcomas [40]. Finally, a study by Mantadakis et al. showed no significant difference in MVD/CD34 between survivors and patients who died with osteosarcoma progression [42]. West et al. showed statistically significant differences between values of MVD/CD31 in histopathologically differentiated subgroups of 42 STSs, but the presence of distant metastases was not correlated with higher values of MVD [37]. A lack of correlation of MVD/CD31 with age, sex, primary tumor site, or a presence of distant metastases in 60 patients with osteosarcomas was also reported [40]. Multiple studies used MVD/factor VIII as an angiogenesis indicator in sarcomas, but no coherent conclusions may be drawn from the studies for this marker [39,43,44,45].

Very little data are available on RMS and MVD and CD105; therefore, our report fills in the gap in this field. In our study, the median CD105-based MVD value in tumor tissue was significantly higher than in the control group, which confirms CD105 expression in immature, proliferating neoplastic vessels. Moreover, we confirmed the prognostic significance of MVD/CD105 (HR = 1.009) and the statistically significant relationship between higher values of MVD/CD31, MVD/CD34, and MVD/CD105, as well as the presence of distant metastases at diagnosis. Di Paolo et al. published an analysis of microvascular density in 30 RMS pediatric cases (18 ERMS and 12 ARMS), wherein they analyzed the expression of CD105, CD31, and VEGF. The authors observed that the expression of the CD105 marker is specifically associated with immature microvessels of the tumor tissue [13]. They also demonstrated the relationship between higher values of the endothelial proliferation indicator (MVD/CD105 and MVD/CD31 expression ratio) and worse prognosis in children with RMS. Moreover, MVD/CD105 was significantly higher in the subgroup of patients with ERMS than in the ARMS group, which was explained by the presence of vascular mimicry in the prognostically worse ARMS, based on alternative forms of vascularization [13]. In general, CD105 expression is high in sarcomas [46]. It may be hypothesized that under the conditions of hypoxia and acidosis found in the environment of sarcoma tumor, the proangiogenic hypoxia-induced factor (HIF-1α) is overexpressed, in turn, amplifying endoglin expression [47]. The determination of the MVD/CD105 rate is considered important when selecting patients for antiangiogenic therapies [48].

Evidence showing a correlation between angiogenesis, tumor growth, and progression has led researchers to study antiangiogenic therapies. To increase the effectiveness of therapy for patients resistant to currently known angiogenesis inhibitors, alternative targets are being studied. Early studies using antiendoglin in antiangiogenic therapy were promising. The most effective antiendoglin drug in development is TRC105, a class-IgG1 chimeric, monoclonal protein, which binds the orphan domain in the extracellular domain of endoglin. TRC105 competitively inhibits the binding of downstream target BMP9, thus preventing the activation of the pathway involving Smad1/5/8 proteins. This results in the maintenance of the resting phenotype of endothelial cells [47]. Furthermore, TRC105 downregulates the expression of VEGF and PDGF [47]. Phase 1 research using a combination of bevacizumab and TRC105 in patients with advanced solid tumors has shown good tolerance and clinical activity of TRC105 in a group of patients resistant to therapy with VEGF inhibitors [49]. A clinical study using a combination of TRC105 and pazopanib in patients with advanced angiosarcomas was recently terminated upon a phase 3 trial interim analysis [13,50], while TCR105 was not tested in the RMS population. Genetic therapies targeting endoglin using siRNA or shRNA are also being undertaken, but further research is still needed to assess the effectiveness of such treatments [47].

There are few limitations that need to be acknowledged. It must be remembered that anti-CD34 antibodies can nonspecifically bind to elements of the tumor stroma, and anti-CD31 may cross-react with morphotic blood elements present in inflamed tumor tissue [48,51]. These two are also panendothelial markers, resulting in a relatively high expression in normal muscle tissue. The conducted Cox regression analysis did not confirm the prognostic value of alveolar histology on death risk. Alveolar histology is a well-defined independent prognostic factor for localized disease but not for metastatic RMS [52]. Thirty-five percent of all RMS patients in this study were children in the fourth stage of the disease, which may have influenced the Cox regression analysis results. Finally, the rarity of rhabdomyosarcomas hinders the enrolment of a large study group.

## 5. Conclusions

The use of angiogenesis markers to assess the degree of tumor vascularization allows better selection of patients for antiangiogenic therapy and enables monitoring of treatment effects [13]. Microvessel density is a widely used quantitative measure of the intensity of angiogenesis. Estimation of MVD makes use of specific markers on the surface of endothelial cells, such as CD31, CD34, factor VIII, or CD105, which allow visualization of the vessel structure in a microscope image [53]. Among them, endoglin, present on activated cells of the vascular endothelium, is considered the most specific marker of immature, small-diameter neoplastic vessels [15]. Endoglin is currently becoming an attractive alternative treatment target in cases resistant to treatment with VEGF inhibitors. The presented results support this trend and confirm the presence of glycoprotein CD105 on the surfaces of endothelial cells in pediatric RMS tumors. Our study suggests the usefulness of MVD/CD105 assessment as a marker of the intensified proliferation of endothelial cells in sarcoma tumors. Despite the relatively small sample size, a relationship between increased microvascular density based on CD105 expression and low rates of overall survival was shown. MVD/CD105 is correlated with unfavorable prognostic factors of RMS survival, such as the alveolar histopathological subtype or the presence of distant metastases at diagnosis. Further research on a large patient population is still needed to confirm the prognostic significance of endoglin in RMS in order to confirm the potential utility of targeted therapy against CD105 as an element of combined RMS treatment.

## Figures and Tables

**Figure 1 jcm-10-00512-f001:**
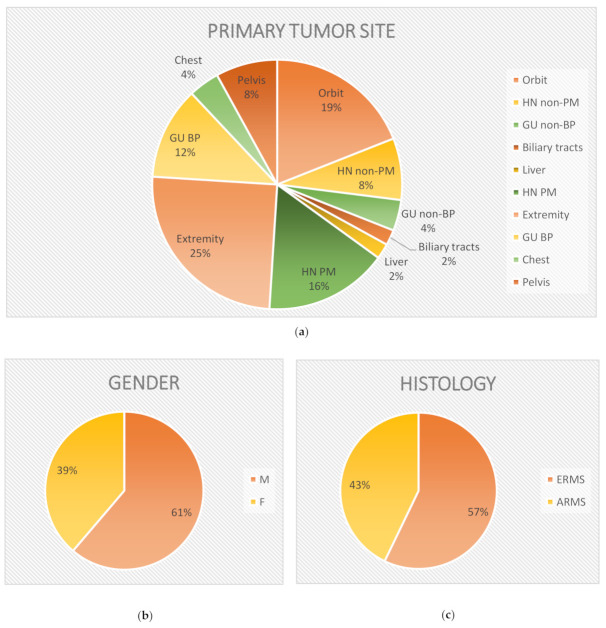
Diversification of study group in terms of primary tumor site location (**a**), sex (**b**), and histopathology (**c**). (**a**) Abbreviations: GU BP, genitourinary tract—bladder and/or prostate; GU non-BP genitourinary tract with the exception of the bladder and prostate; HN PM, head and neck—parameningeal region; HN non-PM, head and neck with the exception of the parameningeal region. (**b**) Abbreviations: M, male; F female.

**Figure 2 jcm-10-00512-f002:**
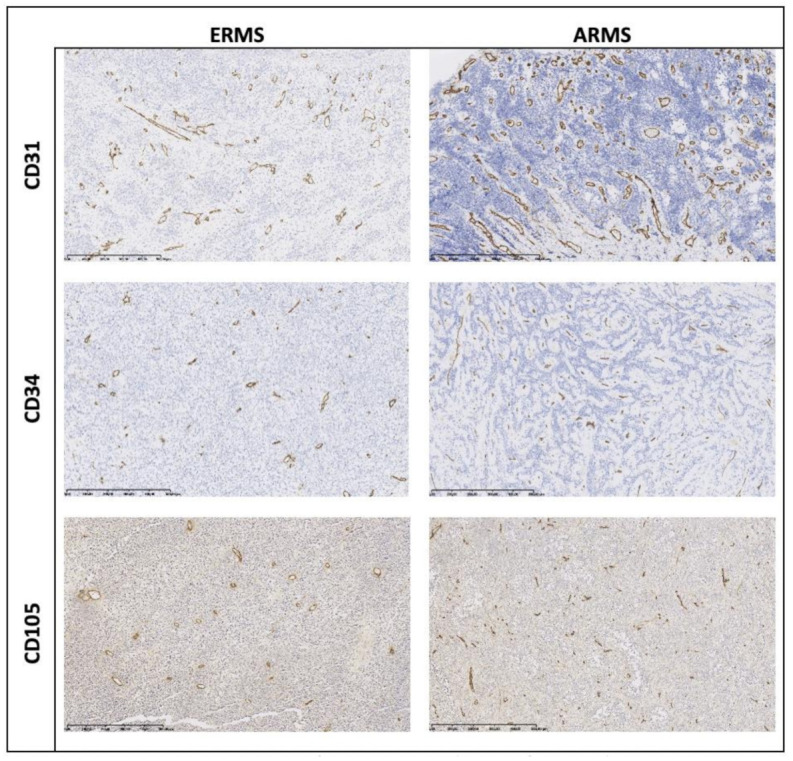
Representative immunostaining for CD31, CD34, and CD105 of ERMS and ARMS. Magnification: 100×.

**Figure 3 jcm-10-00512-f003:**
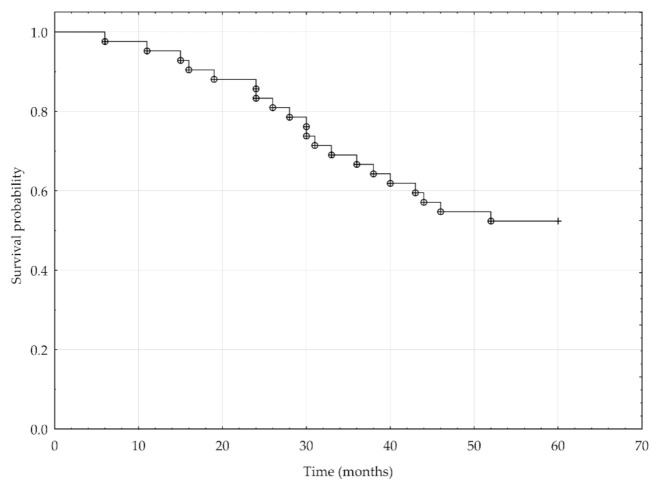
Kaplan–Meier analysis of the 5-year overall survival for 42 RMS patients.

**Figure 4 jcm-10-00512-f004:**
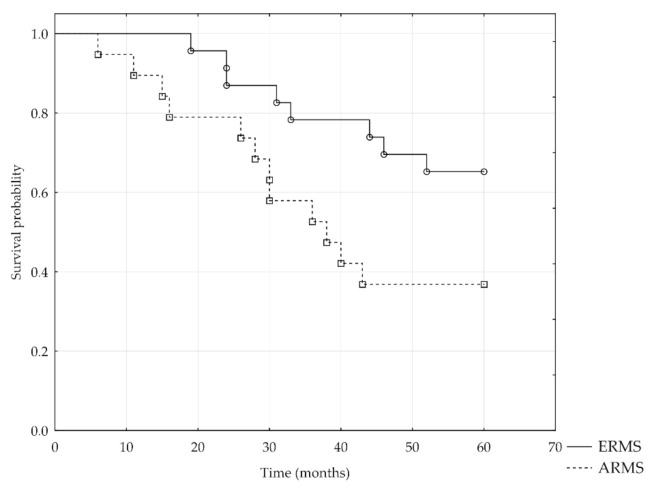
Kaplan–Meier analysis of the 5-year overall survival for patients with ERMS and ARMS.

**Figure 5 jcm-10-00512-f005:**
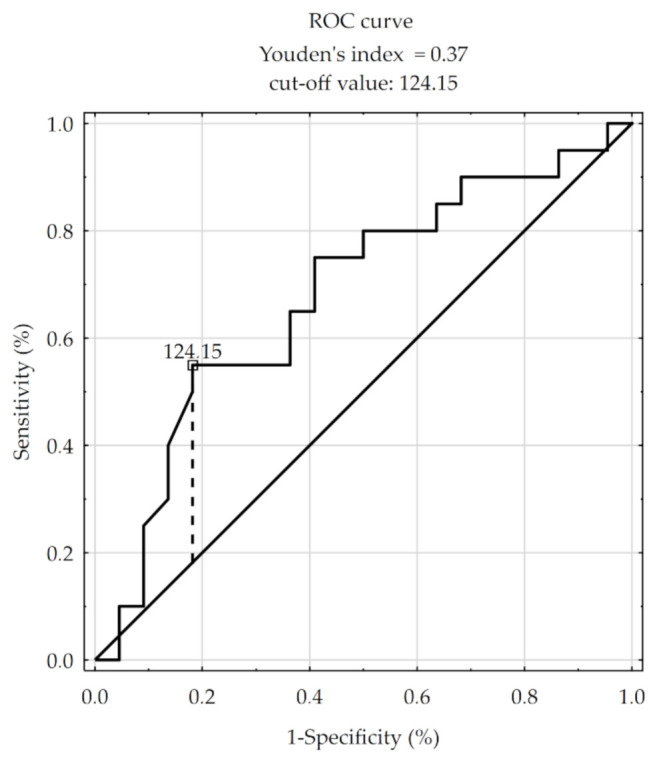
Receiver operating characteristic (ROC) curve for MVD/CD105.

**Figure 6 jcm-10-00512-f006:**
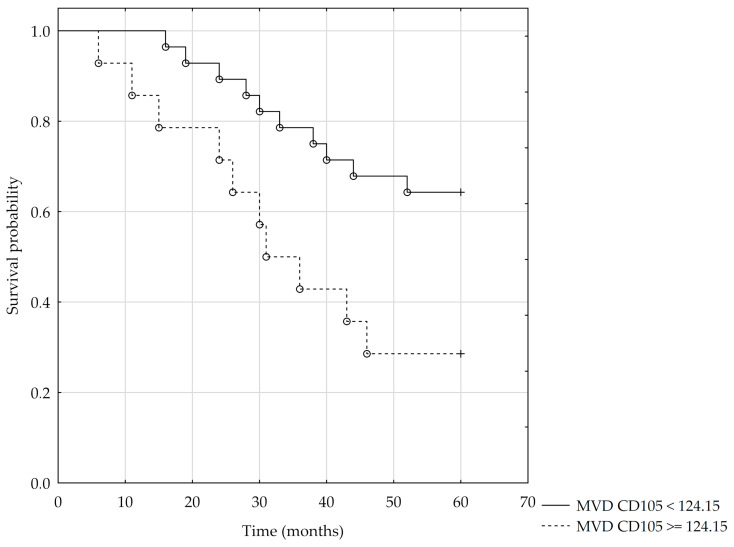
Kaplan–Meier analysis of the 5-year overall survival using the cut-off value = 124.15 for the MVD/CD105.

**Table 1 jcm-10-00512-t001:** Characteristics of 49 participants with rhabdomyosarcoma.

Pt	Age (Years)	Gender	Histology	Primary Site	Primary Size (cm)	T	N	M	TNM Stage	Follow Up	MVD CD31	MVD CD34	MVD CD105
1	2	M	ERMS	Favorable	≤5	1	0	0	1	DOD	56.1	88.4	40.8
2	1 mos	M	ERMS	Favorable	≤5	1	0	0	1	ALV	156.5	156.5	90.1
3	3	M	ERMS	Favorable	≤5	2	0	0	1	TSF	137.8	44.2	22.1
4	6 mos	F	ERMS	Favorable	≤5	1	0	0	1	TSF	236.4	340.1	96.9
5	9 mos	M	ARMS	Unfavorable	>5	2	0	1	4	DOD	335.0	190.5	136.1
6	4	F	ERMS	Unfavorable	>5	2	0	0	3	DOD	363.9	362.2	124.1
7	12	F	ARMS	Unfavorable	>5	2	0	1	4	DOD	161.6	197.3	95.2
8	4	F	ERMS	Unfavorable	>5	2	1	1	4	DOD	137.8	250.0	127.6
9	11	M	ERMS	Favorable	≤5	1	0	0	1	ALV	136.1	154.8	56.1
10	3 mos	F	ARMS	Favorable	≤5	2	0	0	1	DOD	173.5	73.1	125.9
11	9	M	ERMS	Unfavorable	>5	2	0	0	3	ALV	83.3	113.9	76.5
12	4	F	ERMS	Favorable	≤5	2	0	0	1	ALV	132.7	176.9	115.6
13	3	M	ERMS	Unfavorable	>5	2	1	1	4	DOD	236.4	85.0	112.2
14	17	M	ARMS	Unfavorable	>5	2	1	1	4	DOD	309.5	199.0	176.9
15	7	M	ERMS	Unfavorable	>5	2	1	0	3	DOD	124.1	202.4	81.6
16	16	F	ARMS	Favorable	>5	2	0	1	4	DOD	462.6	188.8	200.7
17	5	M	ERMS	Favorable	≤5	1	0	0	1	TSF	91.8	107.1	73.1
18	4	M	ERMS	Unfavorable	>5	2	1	0	3	ALV	15.3	20.4	15.3
19	4	F	ERMS	Favorable	≤5	1	0	0	1	ALV	129.3	107.1	120.7
20	8	F	ARMS	Unfavorable	>5	2	1	1	4	DOD	127.6	142.9	74.8
21	6	F	ERMS	Favorable	>5	2	1	0	1	TSF	219.4	204.1	91.8
22	3	F	ERMS	Unfavorable	>5	1	0	0	3	ALV	54.4	88.4	37.4
23	19 mos	M	ERMS	Unfavorable	≤5	2	0	0	2	ALV	124.1	120.7	68.0
24	2	F	ARMS	Unfavorable	≤5	1	0	0	2	ALV	142.9	176.9	115.6
25	15	M	ERMS	Unfavorable	>5	2	1	1	4	DOD	90.1	209.2	34.0
26	15 mos	M	ARMS	Unfavorable	>5	2	1	1	4	ALV	100.3	79.9	39.1
27	2	M	ARMS	Unfavorable	>5	2	0	0	3	ALV	127.6	243.2	76.5
28	21 mos	M	ERMS	Favorable	>5	2	1	0	1	DOD	161.6	171.8	125.9
29	15	F	ARMS	Unfavorable	>5	2	0	1	4	ALV	132.7	338.4	153.1
30	9	M	ARMS	Favorable	≤5	1	0	0	1	TSF	318.0	88.4	122.4
31	3	F	ERMS	Unfavorable	>5	2	0	0	3	ALV	125.9	102.0	83.3
32	2	M	ARMS	Unfavorable	>5	2	0	1	4	ALV	219.4	363.9	54.4
33	2	M	ERMS	Unfavorable	>5	2	0	0	3	AVL	91.8	56.1	47.6
34	8	M	ERMS	Unfavorable	>5	2	0	1	4	DOD	363.9	511.9	233.0
35	2	M	ERMS	Unfavorable	>5	2	0	0	3	ALV	136.1	136.1	76.5
36	11	F	ERMS	Unfavorable	>5	1	0	1	4	TSF	227.9	227.9	188.8
37	3	M	ERMS	Unfavorable	>5	2	0	0	3	ALV	71.4	112.2	62.9
38	5	M	ARMS	Unfavorable	≤5	1	0	1	4	DOD	353.7	295.9	277.2
39	14 mos	M	ERMS	Unfavorable	>5	2	0	0	3	DOD	212.6	216.0	229.6
40	5	F	ERMS	Unfavorable	>5	2	0	0	3	ALV	205.8	144.6	107.1
41	3	M	ERMS	Favorable	≤5	1	0	0	1	ALV	227.9	263.6	117.3
42	15	F	ARMS	Unfavorable	>5	2	0	0	3	DOD	93.5	132.7	66.3
43	7	M	ARMS	Favorable	≤5	1	0	0	1	ALV	120.7	170.1	125.9
44	2	M	ARMS	Unfavorable	>5	2	0	1	4	DOD	622.4	1052.7	294.2
45	15	M	ARMS	Unfavorable	>5	2	0	1	4	DOD	216.0	251.7	159.9
46	5	M	ERMS	Favorable	≤5	1	0	0	1	TSF	227.9	168.4	108.8
47	8	F	ARMS	Unfavorable	>5	1	0	0	3	DOD	214.3	110.5	100.3
48	5	M	ERMS	Favorable	>5	2	0	1	4	DOD	120.7	127.6	108.8
49	13	F	ARMS	Unfavorable	>5	2	0	0	3	DOD	142.9	159.9	153.1

Abbreviations: Pt, patient; mos, months; M, male; F, female; ERMS, embryonal rhabdomyosarcoma; ARMS, alveolar rhabdomyosarcoma; T1, tumor confined to the anatomic site of origin; T2, extension and/or fixation of the tumor to surrounding tissues/structures; N0, regional lymph nodes not clinically involved; N1, regional lymph nodes clinically involved by neoplasm; M0, no distant metastasis; M1, metastasis present; TNM, pretreatment staging system according to the Intergroup Rhabdomyosarcoma Study; ALV, alive: DOD, died of disease; TSF, too short follow-up and no end-point death.

**Table 2 jcm-10-00512-t002:** Characteristics of the control group.

Pt	Age (Years)	Gender	Surgical Procedure
1	11	F	Tonsillectomy
2	11	F	Tonsillectomy
3	18 mos	F	TGDC excision
4	6	F	TGDC excision
5	5	M	TGDC excision
6	8	M	TGDC excision
7	15	M	TGDC excision
8	15	F	Tonsillectomy
9	15	F	Tonsillectomy
10	6	M	Tonsillectomy
11	15	F	Tonsillectomy
12	17	F	Tonsillectomy
13	17	F	Tonsillectomy
14	13	F	Tonsillectomy
15	17	F	Tonsillectomy
16	8	M	Tonsillectomy
17	15	M	Tonsillectomy
18	15	F	Tonsillectomy

Abbreviations: Pt, patient; mos, months; M, male; F, female; TGDC, thyroglossal duct cyst.

**Table 3 jcm-10-00512-t003:** Microvessel density (MVD) evaluation based on the expression of selected endothelial markers in the study group and in the control group (Mann–Whitney U Test).

Variables	Study Group			Control Group			Mann-Whitney U Test (Z)	*p*-Value
	Mean	SD	Median	Mean	SD	Median		
MVD/CD31	185.17	112.69	142.86	352.04	186.40	322.28	−3.68	*p* < 0.001
MVD/CD34	194.40	156.83	168.37	711.73	205.48	770.41	−5.83	*p* < 0.001
MVD/CD105	110.65	61.36	107.14	74.07	107.52	9.35	2.75	*p* = 0.006

**Table 4 jcm-10-00512-t004:** Cox regression analysis.

Cox Regression Analysis—5 Years. Model Chi-Square = 22.25, *p* = 0.0002				
Parameter	Parameter Estimate	Chi-Square	*p*-Value	Hazard Ratio (HR)
Age < 1	2.07	5.75	0.016	7.97
Age ≥ 10	1.82	11.07	0.001	6.22
N1	2.44	13.65	0.0002	11.51
MVD/CD105	0.01	6.19	0.013	1.009

## Data Availability

Data are available upon request to the corresponding author.

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
