# Peer review of "Endoglin Expression and Microvessel Density as Prognostic Factors in Pediatric Rhabdomyosarcoma"

_jcm, 2021, doi:10.3390/jcm10030512_

Round 1
Reviewer 1 Report
In this paper the authors analyze the MVD in RMS tumor tissue, using CD31, CD34 and CD105 as markers and then they correlate the value obtained with clinical and pathomorphological parameters of patients.
In my opinion, the research present numerous issues. First of all the research was not novel, in fact there is another paper published in 2018 (that was also cited by the authors) in which the prognostic value of endoglin in rhabdomyosarcoma has already been reported with the same method used in this paper.
Moreover the authors in the study population section talk about control tissues but the number and the clinical characteristics are not specified in the text. In addition the exact role of this control tissues for the experimental purpose is not well defined in the text.
Figure 2 please specify in the caption if the tumor is ARMS or ERMS. Not generic RMS.
Figure 3 is significant??
I suggest to report also the Kaplan Meyer Curve as figure.
Minor comments please adjust the abbreviation
Author Response
Dear Reviewer
Thank you for all your comments on our manuscript. We have worked on correcting the paper according to all suggestions.
Kind regards, Wojciech Kukwa.

Reviewer 2 Report
Manuscript Number: JCM 1016022
Title: Endoglin expression and microvessel density as 2 prognostic factors in pediatric rhabdomyosarcoma
Article Type: Original paper
In this article the authors assessed the microvessel density (MDV)in pediatric rhabdomyosarcoma(RMS) by analyzing the expression of angiogenesis markers (CD31, CD34, CD105) in order to study its prognostic role in such an aggressive neoplasm. They observed the values of MVC/CD105 was significantly higher in alveolar RMS (ARMS) and RMS with metastatic disease, correlating also with a higher risk of death. This is an important result as it emphasize how MDV/CD105 could be utilized as a prognostic markers and is able to stratify RMS patients with a worse overall survival. Moreover, CD105 (endoglin) is also been reported as an anti-angiogenesis treatment for soft-tissue sarcoma, making an appealing druggable target for pediatric RMS.
The study is extremely interesting but there are several major issues that need to be addressed to improve the quality of the manuscript. The manuscript does not follow the usual format for an original paper. The introduction is way too long and it should not be longer than a page. It also needs to be more focused and clear, in particular the clinical need of the study has to stressed better. What should be emphasized in the introduction is the reason you decided to study MVD in rhabdomyosarcoma. All the rest is interesting but not aimed for article purposes. The same argument goes for the discussion: too lengthy and confusing, instead should be shorter and to the point.
There are also several minor comments:
- The control group should be explained better: how many cases? What characteristics? Perhaps a table in the supplementary material could help.
- Throughout the Results Section at times the authors uses the term ‘sarcoma cases’ instead of RMS. This is confusing: the cohort studied is represented by RMS samples. Sarcoma is too generic and implies many different histotypes.
- Were Alveolar RMS cases were investigated for PAX3/7 FOXO1 status?
- Table 1: Age is usually reported in years. Also, the ‘primary site’ is reported as favorable/unfavorable which is fine but it should be indicated and referenced according to what criteria. IRS?
- The overall survival is analyzed over 42 RMS cases: it should be explained why only 42 instead of 49.
Author Response

(The authors gave the same response as above.)

Round 2
Reviewer 1 Report
The authors answered all the reviewers' questions
Author Response
Dear Reviewer,
We thank you again for reviewing our manuscript. We only shortened the Discussion according to Reviewer #2 comment and added a "Limitations of the Study" paragraph.
Kind regards, Wojciech Kukwa.
Reviewer 2 Report
The new version of the manuscript is much improved, howver the discussion section is yet too lengthy. If it could be slightly condensed I believe it would be much better.
Author Response
Dear Reviewer,
We thank you again for reviewing our manuscript. We only shortened the Discussion according to your comment and added a "Limitations of the Study" paragraph.
Kind regards, Wojciech Kukwa.